# With or without a Tourniquet? A Comparative Study on Total Knee Replacement Surgery in Patients without Comorbidities

**DOI:** 10.3390/medicina59071196

**Published:** 2023-06-25

**Authors:** Mehmet Albayrak, Fatih Ugur

**Affiliations:** 1Department of Orthopaedics and Traumatology, Ozel Tekirdag Yasam Hospital, 59030 Tekirdag, Turkey; 2Department of Physiotherapy, Vocational School of Health Services, Istanbul Rumeli University, 34750 Istanbul, Turkey; 3Department of Orthopaedics and Traumatology, School of Medicine, Kastamonu University, 37150 Kastamonu, Turkey; fugur.md@gmail.com

**Keywords:** comorbidities, total knee arthroplasty, tourniquet, pain

## Abstract

*Background and Objectives:* This study aimed to determine the effects of tourniquet use and the complications of total knee arthroplasty (TKA) in patients without comorbidities to investigate whether tourniquet application can be employed without adverse effects and to assess its impact on the occurrence of any complications. *Materials and Methods:* A total of 106 patients who underwent unilateral TKA were divided randomly into two groups according to whether a tourniquet was used during the surgery or not. Patients with comorbidities (except arterial hypertension) were excluded from the study. Knee Injury and Osteoarthritis Outcome Score, joint range of motion, visual analog scale (VAS) score, total blood loss during and after surgery, postoperative analgesic consumption, and side effects were the main factors evaluated in the study. *Results:* In the tourniquet group, where the VAS scores were higher, the use of analgesics was also significantly higher. While there was no statistically significant difference in total blood loss between the tourniquet and non-tourniquet groups, the postoperative and occult blood losses were higher in the tourniquet group. The differences between the two groups in all other parameters were very small and not statistically significant. *Conclusions:* The findings of the current study suggest that when the comorbidities of patients are thoroughly documented and clarified prior to surgery, tourniquets should be applied selectively to individuals without any pre-existing health conditions.

## 1. Introduction

Total knee arthroplasty (TKA) is currently the most effective treatment option for managing pain, movement restriction, and deformity in advanced-stage knee osteoarthritis and is one of the most common orthopedic surgeries performed worldwide [1,2]. It is predicted that by 2030, the number of TKA operations performed in the USA every year will have reached 1.26 million, an approximately 80% increase from the present [3].

With the gradual modifications in prosthetic design over the years and the development and availability of more anatomical prostheses, greater surgical success can now be obtained with refinements in the surgical technique [4]. An important component in the evolution of TKA might be the role of tourniquet use during the surgery. Tourniquet application ensures a non-hemorrhagic and bloodless surgical field by impeding intraoperative blood flow to the operating field, which can improve visualization. To achieve optimal results and enhance the long-term survival of the prosthesis without loosening, cement application on the sections of the prosthesis’ surface that come into contact with the bone should be performed under a tourniquet during hardening, as recommended by multiple researchers [5,6]. When applied, a tourniquet subjects the extremities to significant mechanical pressure and compresses the underlying skin, muscles, nerves, and vessels; therefore, all parts of the limb experience ischemia [7]. Despite its well-established advantages, using a tourniquet while performing TKA can result in postoperative pain or hyperalgesia in the thigh and leg, wound site and surrounding infections, thromboembolic events, nerve damage, formation of ischemia-related metabolites resulting in rhabdomyolysis, skin problems such as fat necrosis in soft tissues, and patellofemoral maltracking by affecting the quadriceps’ mobility during surgery [8,9,10,11,12,13]. While concerns have been raised about potential complications of tourniquet use during TKA, there is currently limited evidence to support these concerns, especially in patients without a peripheral arterial disease, but because the patients undergoing TKA are generally over 65 years of age, arterial diseases that can pose risks are present in the vast majority of patients. Therefore, if comorbidities are present, using a tourniquet while performing TKA can increase the risks. Nonetheless, the American Association of Hip and Knee Surgeons has reported that approximately 95% of orthopedic surgeons use tourniquets in some form during TKA [11].

On the other hand, some researchers (e.g., Husted et al.) claim that the benefits of using a tourniquet during TKA is a myth because tourniquet use does not really provide any advantage during TKA and is even disadvantageous [14].

There is currently no consensus on the benefits of using a tourniquet during TKA; the primary aim of this study was to bridge the existing knowledge gap and provide insights into the unresolved aspects of this field. In this study, we hypothesized that the use of tourniquets in patients without comorbidities is beneficial for the long-term survival of TKA. Additionally, we investigated whether the use of a tourniquet during TKA is beneficial in patients without comorbidities. In particular, we aimed to determine the benefits and possible side effects of tourniquet use for TKA on perioperative blood loss and postoperative knee range of motion (ROM), pain, and complications in patients without comorbidities.

## 2. Materials and Methods

A retrospective cohort study design was planned to include patients over 65 years of age who had undergone unilateral TKA for primary knee osteoarthritis between February 2011 and September 2015 in Tekirdağ Yasam Hospital Orthopedics and Traumatology Clinic, Tekirdağ, Turkey. Ethical approval for the present study was obtained from the Clinical Research Ethics Committee, Faculty of Medicine, Namık Kemal University, Tekirdağ, Turkey (approval no.: 2021.114.04.09). All the patient data from the hospital’s medical records were obtained for this retrospective study without extracting personal identification details. The study was conducted in accordance with the Declaration of Helsinki; all the patients provided written informed consent to participate in the study and to the use of their data for scientific purposes by contacting them again.

The indications for TKA included patients’ non-response to conservative treatments for primary knee osteoarthritis (drugs, physical therapy, weight loss, etc.) [15], presence of severe pain, and Kellgren–Lawrence grades III and IV [16] (narrowing of the joint space, presence of subchondral bone cysts, bone sclerosis, and osteophyte formation). The study exclusion criteria consisted of the following comorbidities: previous surgery or treatment related to tumors or tumor-like formations, cerebrovascular disease, liver disease associated with liver dysfunction, chronic obstructive pulmonary disease, heart disease, peripheral vascular pathology, coagulation disorder, history of thromboembolic events, diabetes mellitus, class II or III obesity (body mass index (BMI) ≥ 35 kg/m^2^), asthma, iron deficiency anemia, hyperlipidemia, diagnosis of chronic obstructive pulmonary disease, and history of narcolepsy. Primary hypertension controlled with antihypertensive therapy was not considered an exclusion criterion. We also excluded patients with active infections and those currently using steroids.

A total of 144 patients were screened for inclusion in the study; among them, 21 refused to participate or could not be reached, and 17 were excluded due to the presence of comorbidities. Based on the findings obtained from this study, when a difference of 150 units between the means and standard deviations of 200 and 300, respectively, was considered, the statistical power of the study was determined to be 85% at a 95% confidence level and a significance level of 0.05. The study included a sample size of n1 = n2 = 52, with a total sample of *n* = 104 individuals. Finally, 106 patients who met the inclusion criteria and did not have any of the exclusion criteria were included in the study (Figure 1).

The patients were further randomly divided into two groups (*n* = 53 patients each) based on tourniquet usage during surgery. The first group (the tourniquet group) included patients on whom TKA was performed with a tourniquet inflated before incision and deflated after wound closure. The second group (the non-tourniquet group) included patients on whom TKA was performed without the use of a tourniquet. 

### 2.1. Operation Procedure

All the surgeries were performed by the hospital’s same senior orthopedic surgeon in a laminar flow-embedded operating room with a standard orthopedic operating table. After spinal anesthesia application, the patient was placed in a supine position. In the tourniquet group, the tourniquet was inflated during knee flexion (45° flexion). After the application of the Esmarch bandage, the pressure was balanced (100 mmHg higher than the patient’s synchronous systolic blood pressure), and after wound closure, the tourniquet was depressed. After sterile dressing and draping, the surgery began with a 10 to 12 cm anterior midline skin incision, followed by a medial parapatellar capsule incision. After resection of the osteophytes and soft tissue debridement, the knee was flexed as much as possible, preventing the patellar tendon from being ruptured iatrogenically. After bony resections of both the femur and tibia using gap-balancing techniques, both bones were positioned perpendicular to the mechanical axis. None of the patellae resurfaced, but all the patellar surfaces underwent patellar resection arthroplasty with electrocautery. The implants used were posterior cruciate ligament-sacrifying and inserts were mobile (NexGen, Complete Knee Solution, Zimmer, Warsaw, IN, USA). All the implants were cemented. Prior to their cementation, the bone–implant interfaces were dried thoroughly and carefully with a gauze and suction in both groups. After the adaption of the implants, some time was spent waiting for the cement to become rigid, and the layers were closed in the reverse order of their opening, but at the same position (45° knee flexion) to remove the tension when closed in full extension. In both groups, a drain was used and removed after 24 h. In the tourniquet group, the tourniquets were deflated after skin closure.

The same intraoperative and postoperative protocols were followed for administering spinal anesthesia, anti-inflammatory drugs, and opioids, and for postoperative pain control and the use of antithrombotic and antibiotic prophylaxis agents.

Low-molecular-weight heparin treatment was started 12 h postoperatively and continued for 30 days. Antibiotic prophylaxis was started with a first-generation cephalosporin (4 × 1 g) 1 h before the operation and continued for the next 24 h. Postoperative rehabilitation comprising passive ROM on the continuous passive motion device was started on the first postoperative day, and the patient was mobilized and allowed to engage in full-weight-bearing activities with the help of a walker on the same day. Ladder exercises were started on the second postoperative day. All the exercises were continued until the patient’s discharge, and after discharge, walking and ladder exercises were continued up to the 8th week.

### 2.2. Measurement and Evaluation of Data

The following demographic and clinical data were noted for all the patients: age, gender, BMI, Knee Injury and Osteoarthritis Outcome Score (KOOS), preoperative and postoperative joint ROM, visual analog scale (VAS) score, blood loss during and after surgery, postoperative analgesic consumption, and vascular, neurologic, and metabolic complications which can impair the extremity or the patient itself. KOOS is a self-administered, self-explanatory 42-item questionnaire that covers five patient-relevant dimensions: pain, other disease-specific symptoms, average daily function, sport and recreation function, and knee-related quality of life. KOOS scores (from 0–100) were calculated for all the patients preoperatively and at the end of the 8th postoperative week. A score of 0 indicated serious knee problems, and a score of 100 indicated no knee problems [17].

The knee ROM was measured with the help of a universal goniometer. VAS is a 10 cm line used to measure the pain felt by a patient on a scale of 0–100 [18]. It was used in the present study to measure the minimum and maximum pain felt by the patients during movement at every walking exercise time. Both the knee ROM and VAS scores were assessed preoperatively, on the first three postoperative days, and at the end of the 8th postoperative week.

The hemoglobin and hematocrit values were recorded preoperatively and on the first two days after surgery. The total blood loss was calculated using the Nadler approach [19,20]. The intraoperative blood loss was calculated by subtracting the volume of serum used for cleaning the surgical field during surgery from the total amount of fluid in the aspirator cup and adding the remaining value to the value obtained from the gauze visual analog [21]. The postoperative blood loss was calculated as the volume in the drain system [22]. The occult blood loss was calculated by subtracting the sum of the intraoperative and postoperative total blood losses from the overall blood loss.

The analgesic doses and time intervals for each drug were also recorded. An oral nonsteroidal anti-inflammatory drug (etodolac 70 mg/day) was routinely prescribed for pain control. If it was insufficient, an intramuscular injection of tramadol HCl (100 mg/day) was used. Intramuscular injection of pethidine HCl (100 mg/day) was prescribed to some patients whose pain did not improve with tramadol HCl.

Although TKA is commonly performed, it is a major surgical procedure and is prone to complications such as infection, loosening, and the need for reoperation [23]. The patients were evaluated according to the complications that occurred.

## 3. Statistical Analysis

Estimated power was 0.80, alpha (margin of error): 0.05, effect size was 0.4. Accordingly, the sample size was determined as 106 for the chi-square test. All files (106 files) were included in the study, as the number of files remaining after assessing all the files according to the exclusion criteria. The data distribution was examined using the Shapiro–Wilk test. Comparisons between two independent groups with normal distribution were made using the independent sample t-test, while comparisons between two independent groups without normal distribution were made using the Mann–Whitney U test. When the normal distribution assumption was not met for comparisons between two dependent groups, the Wilcoxon test was used, and when the assumption was not met for more than two dependent groups, the Friedman test was used. Descriptive statistics of the data were presented as mean ± standard deviation or median (min–max). All the statistical analyses were performed using the IBM SPSS Statistics 26.0 software and reported at a significance level of α = 0.05.

## 4. Results

A total of 106 patients (94 female, 12 male; mean age: 69 years old; mean BMI: 26.8 kg/m^2^) were included in the study and two groups were formed. The two study groups were statistically comparable in terms of the patients’ ages and BMI. Additionally, as shown in Table 1, there were no statistically significant differences between the tourniquet and non-tourniquet groups in terms of the operation time (59 ± 14 min and 62 ± 11 min, respectively) which is also the tourniquet time for the tourniquet group because the tourniquets were deflated after skin closure and the preoperative hemoglobin and hematocrit values (*p* > 0.05).

### 4.1. Knee Injury and Osteoarthritis Outcome Score

The changes in the KOOS values from the preoperative period in the two groups were compared. As shown in Table 2, there was no significant difference between the two groups in any of the KOOS parameters before and after the surgery (*p* > 0.05).

### 4.2. Knee Range of Motion

The changes in the ROM values from the preoperative period in the tourniquet and non-tourniquet groups were compared. There was no significant difference between the two groups in terms of knee ROM before and after surgery (Table 3).

### 4.3. Visual Analog Scale Scores

When the changes in the VAS scores from the preoperative period in the tourniquet and non-tourniquet groups were compared, the changes from the preoperative period to the 1st and 2nd postoperative days and 8th postoperative week were found to be significantly smaller in the non-tourniquet group, while there was no significant difference between the changes from the preoperative period to the 3rd postoperative day in the two groups (Table 4).

### 4.4. Blood Loss

The postoperative and occult blood losses in the tourniquet and non-tourniquet groups were compared. According to the results shown in Table 5, the postoperative and occult blood loss values were significantly higher in the tourniquet group than in the non-tourniquet group. However, there was no significant difference between the two groups in terms of total blood loss and transfusion values (*p* > 0.05).

### 4.5. Analgesic Consumption

The analgesic doses used by the tourniquet and non-tourniquet groups were compared. According to the results shown in Table 6, there were statistically significant differences between the two groups in terms of the doses of the etodolac, tramadol, and pethidine analgesics used, and the doses of all three analgesics were found to be significantly higher in the tourniquet group than in the non-tourniquet group (*p* < 0.05).

**Table 5 medicina-59-01196-t005:** Comparison of blood losses in the tourniquet and non-tourniquet groups (mL).

Blood Loss	Non-Tourniquet Group (*n* = 53)	Tourniquet Group (*n* = 53)	*p*-Value *
Total blood loss	850 (750–1100)	880 (700–1110)	0.799
Intraoperative blood loss	400 (150–540)	-	NA
Postoperative blood loss	250 (200–450)	400 (300–650)	<0.001
Occult blood loss	200 (10–550)	480 (200–750)	<0.001
Transfusion	1 (1–1)	1 (1–1)	1

* Mann–Whitney U test. The data are presented as median (min–max). NA: not applicable.

### 4.6. Complications

All the complications observed during hospitalization and subsequent follow-up at the end of the 2nd and 8th postoperative weeks were evaluated by the surgeon. No patients in either group exhibited postoperative venous thromboembolism, pulmonary embolism, neurovascular injury, or wound complications such as skin bruising clinically or metabolic disturbance due to the body’s fluid balance regulated through fluid intake and excretion and with blood tests. Five patients from the tourniquet group and four patients from the non-tourniquet group developed transient orthostatic hypotension on the 1st postoperative day while starting to walk.

## 5. Discussion

The main findings of our study indicate that tourniquet application can be safely utilized in patients without comorbidities, as it does not lead to complications. Furthermore, the use of a tourniquet does not pose significant challenges compared to non-use, as it only results in easily manageable postoperative pain for 1–2 days without causing any additional distress. Over the years, an increasing number of studies have concluded that TKA without tourniquet use is superior to TKA with tourniquet use. Moreover, many studies have pointed out that the rates of thromboembolic events, wound infections, and other complications are lower when TKA is performed without a tourniquet [7,8,24,25,26,27]. Nevertheless, there is currently no definitive evidence supporting the termination of tourniquet use in extremity surgeries [12,28,29]. Previous studies have reported variable complication rates related to tourniquet use in TKA [7,9]. The most important reason for this discrepancy is that the previous studies were not rigid enough to exclude patients with comorbidities that could affect surgical outcomes [7,11]. The principal outcome of the present study was the determination of any complications if we use a tourniquet during TKA in patients suffering from knee arthrosis who have no underlying comorbidities. No complications related to tourniquet use in TKA were observed in these patients.

The most serious and potentially preventable complications in total knee prosthesis applications are deep vein thrombosis (DVT) and pulmonary embolism. However, no studies have indicated that the use of a tourniquet for such applications increases these risks [11,26]. Gazendam et al. [27] stated that high rates of thromboembolism, one of the foremost postoperative complications, are not sufficient to methodologically evaluate surgical outcomes, and also stated that due to insufficient data analysis on whether tourniquets or comorbidities cause complications in patients that it is unknown. Consequently, it cannot be conclusively stated whether the development of thromboembolism is related to the use of tourniquets [27]. In a study by Fukuda et al. [30], TKA alone was not found to be a risk factor for DVT, and no statistically significant increase was found in the incidence of DVT in patients undergoing TKA with or without a tourniquet. There are multiple risk factors for DVT; for instance, regardless of the type of surgery, patients with even class II or III obesity are two times likelier to develop it than non-obese patients [31]. Parvizi et al. [7] concluded that in the context of TKA, the application of a tourniquet should be regarded as the accepted standard of care, except in cases where a preoperative vascular evaluation clearly indicates that its use would be unsuitable. In another study, Parvizi et al. [32] mentioned that the presence of hypercoagulability, metastatic cancer, stroke, sepsis, or chronic obstructive pulmonary disease increases the risk of venous thromboembolism (VTE) by 3%. Based on this, they argued that VTE risk should be assessed for each person [32]. No cases of DVT or pulmonary embolism were found in the present study, whose participants did not have comorbidities.

In the present study, the VAS scores decreased daily over the first three postoperative days, and there was significantly more pain in the tourniquet group during their hospital stay than in the non-tourniquet group. Thus, the non-tourniquet group exhibited reduced analgesic requirements during the initial postoperative days. While this concurs with the findings of some other studies [33,34,35], a few others have reported conflicting results [25,26]. Danoff et al. [34] noted that the significant differences in VAS scores in their study did not indicate a clinically significant change. Moreover, Wylede et al. [33] evaluated the importance of chronic pain after TKA and reported that pain 48 h after surgery should not be considered chronic pain, which is persistent pain three months after surgery. In addition, factors covering the perioperative period (e.g., anxiety) and those specific to the postoperative recovery period (e.g., acute postoperative pain) have the potential to affect functional outcomes [33]. Notably, in the present study, the three months required to determine the presence of chronic pain in patients exceeds the follow-up duration of our publication; therefore, this assessment was not included. The existing literature presents varying KOOS values with respect to the outcomes of tourniquet use. While Ejaz et al. found better KOOS values with tourniquet use [32], Teitsma et al. [28] found results with minimal differences. In the present study, the overall KOOS values of the two groups were not statistically different; they showed only a minimal difference.

There are also diverse postoperative ROM values in the literature for patients who underwent TKA with or without a tourniquet. Ejaz et al. [35] and Li et al. [36] reported better ROM results in the non-tourniquet groups in their studies, especially in the first days of the postoperative period. In the present study, there was no significant difference between the two groups in terms of ROM improvement after surgery.

In the studies by Smith et al. [37] and Zhang et al. [38], intraoperative bleeding was reduced with tourniquet use, but tourniquet use had no additional benefits regarding postoperative bleeding, total blood loss, or transfusion rate. In the present study, postoperative and occult blood losses were more common in the tourniquet group, but the postoperative transfusion rate and total blood loss in the two groups were not statistically different. According to Tetro et al. [39], tourniquet use is not effective in reducing the overall blood loss volume, which was also shown by the meta-analyses conducted by Tai et al. [8] and Smith et al. [37].

In total knee prosthesis applications, it is known that tourniquet use reduces the metabolic stress response. The stress response, which is related to excessive postoperative inflammation and muscle damage which is measured with CRP and creatine phosphokinase levels, has been observed to decrease with the use of a tourniquet [13,40]. In a study that investigated effective cementation and tourniquet use, better tibial component cementation was achieved during tourniquet-assisted surgeries than during surgeries performed without a tourniquet [41]. Hedge et al. [42] reported that a bloodless surface is particularly effective in cementation, supporting the use of a tourniquet. Although we did not compare cementation between the tourniquet and non-tourniquet groups in the present study, some publications report that cementation performed with a tourniquet is better than that without a tourniquet, while others suggest that achieving the same level of effectiveness in cementation is possible without the use of a tourniquet, provided that a dry environment during cementation is ensured. Nonetheless, the use of a tourniquet should be preferred during total joint replacement surgeries as it provides a bloodless environment during the operation and cementation stage.

This study was not without limitations. First, outcome measurement was continued for only 8 weeks postoperatively; however, as the aim of the present study was to investigate the effects of tourniquet use on early recovery after TKA, we deemed this period to be sufficiently inclusive. Second, tranexamic acid, which is commonly used in current arthroplasty practice was not included in the present study because it was not used in any of the patients in this study. Third, preoperative psychological evaluations were not conducted for the patients in this study, which is a deficiency in terms of postoperative pain management. Fourth, the data are nearly one decade old and some developments took place in surgical techniques. Finally, the sample size was small and insufficient to observe an adequate number of complications, and the study participants were from a single orthopedic center. Further multicentric studies with a greater number of patients are needed to confirm the findings of the present study.

## 6. Conclusions

Although tourniquet usage may not seem to provide significant advantages compared to non-usage, and even though it can be easily managed during the postoperative period, it can cause a temporary extremity pain for 1 to 2 days, which may reduce patient comfort and, additionally, while the total blood loss during the surgical process and postoperative period may not differ significantly between the two groups, the presence of comorbidities in patients remains crucial. The most important point here is the comorbidities of the patient. As the increase in the long-term survival of prostheses is closely associated with the cementing process being performed in a bloodless environment as indicated in various studies, and considering that tourniquet usage does not lead to complications, particularly in patients without comorbidities, it can be beneficial in avoiding second surgeries due to prosthesis loosening, thereby preventing work loss and expenses. The results of the present study indicate that tourniquets should be used for patients without comorbidities when the comorbidities are well documented and clarified before the surgery.

## Figures and Tables

**Figure 1 medicina-59-01196-f001:**
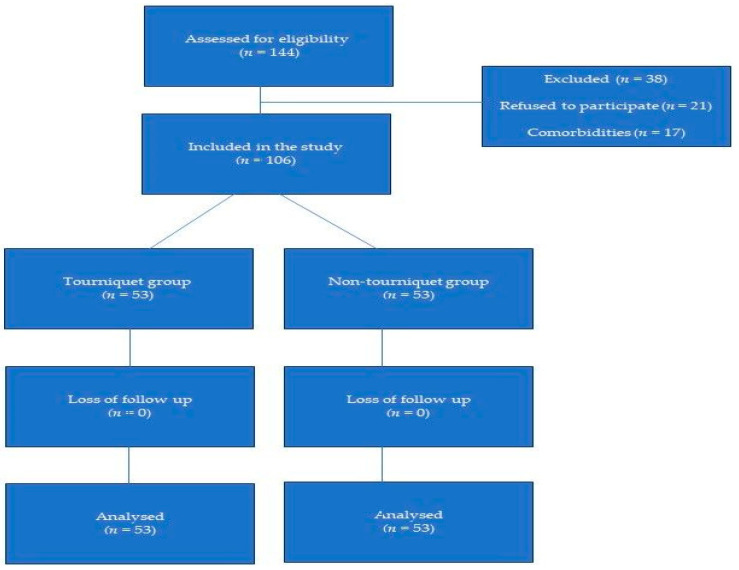
Flowchart of the patients.

**Table 1 medicina-59-01196-t001:** Comparison of the demographic data of the tourniquet and non-tourniquet groups.

	Non-Tourniquet Group (*n* = 52)	Tourniquet Group (*n* = 52)	*p*-Value *
Age	72 (61–82)	71 (60–81)	0.546
Height (cm)	168 (158–178)	167 (158–179)	0.980
Weight (kg)	75 (59–92)	78 (68–92)	0.224
Hematocrit (%)	43 (36–48)	42 (36–48)	0.290
Hemoglobin (g/dl)	13 (11–15)	13 (12–15)	0.222
Operation time (min)	61 (45–75)	59 (45–70)	0.216

* Mann–Whitney U test. The data are presented as median (min–max).

**Table 2 medicina-59-01196-t002:** Comparison of the changes in the Knee Injury and Osteoarthritis Outcome Score (KOOS) values from the preoperative period in the tourniquet and non-tourniquet groups.

Non-Tourniquet Group				Tourniquet Group			*p*-Value *
	Preop	Postop	Δ_diff_	Preop	Postop	Δ_diff_	
KOOS							
Symptom	60 (34–74)	67 (55–79)	8 (−17:40)	53 (37–74)	63 (52–78)	9 (−20:37)	0.711
Pain	49 (32–70)	71 (53–87)	20 (−12:54)	55 (42–68)	70 (52–84)	13 (−7:37)	0.056
ADL	55 (40–74)	73 (61–87)	19 (−10:41)	60 (44–70)	73 (58–90)	14 (−9:44)	0.388
Sports	23 (11–43)	39 (17–58)	17 (−17:42)	29 (14–40)	33 (15–55)	7 (−21:34)	0.101
Qol	30 (12–52)	50 (33–68)	23 (−9:52)	26 (10–47)	62 (52–70)	36 (9:58)	0.062

* Mann–Whitney U test. Preop: preoperative; Postop: postoperative; Δ_diff_: difference between preoperative and postoperative values. The data are presented as median (min–max). Qol: quality of life. ADL: function in daily living.

**Table 3 medicina-59-01196-t003:** Comparison of the changes in the range of motion (ROM) values from the preoperative period in the tourniquet and non-tourniquet groups (degrees).

	Non-Tourniquet Group	Tourniquet Group	*p*-Value *
ROM			
ROMΔ1	−53.54 ± 12.24	−55.71 ± 14.93	0.415
ROMΔ2	−46.13 ± 11.25	−54.86 ± 14.55	0.086
ROMΔ3	−39.37 ± 11.04	−40.71 ± 12.19	0.555
ROMΔ8	−8.47 ± 9.11	−13.15 ± 13.90	0.106

ROMΔ1: change in ROM from preoperative to postoperative day 1; ROMΔ2: change in ROM from preoperative to postoperative day 2; ROMΔ3: change in ROM from preoperative to postoperative day 3; ROMΔ8: change in ROM from preoperative to postoperative week 8. * Independent samples *t*-test. The data are presented as median (min–max) or mean ± standard deviation.

**Table 4 medicina-59-01196-t004:** Comparison of the changes in visual analog scale (VAS) scores from the preoperative period in the tourniquet and non-tourniquet groups.

	Non-Tourniquet Group	Tourniquet Group	*p*-Value *
**VAS scores**			
VASΔ1	−2 (−5:1)	0 (−3:3)	<0.001
VASΔ2	−3 (−6:−1)	−2 (−5:1)	<0.001
VASΔ3	−4 (−6:−2)	−4 (−7:−1)	0.776
VASΔ8	−8 (−10:−5)	−7 (−10:−4)	0.035

VASΔ1: change in VAS score from preoperative to postoperative day 1; VASΔ2: change in VAS score from preoperative to postoperative day 2; VASΔ3: change in VAS score from preoperative to postoperative day 3; VASΔ8: change in VAS score from preoperative to postoperative week 8. * Mann–Whitney U test. The data are presented as median (min–max) or mean ± standard deviation.

**Table 6 medicina-59-01196-t006:** Comparison of the daily doses of analgesics used by the tourniquet and non-tourniquet groups (mg).

Analgesic	Non-Tourniquet Group (*n* = 53)	Tourniquet Group (*n* = 53)	*p*-Value *
Etodolac	40 (35–45)	60 (45–85)	<0.001
Tramadol	50 (30–80)	80 (40–100)	<0.001
Petidine	20 (4–50)	50 (40–100)	<0.001

* Mann–Whitney U test. The data are presented as median (min–max).

## Data Availability

Not applicable. The data and materials generated/analyzed in the present study are available from the corresponding author upon request.

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
