# Peer review of "With or without a Tourniquet? A Comparative Study on Total Knee Replacement Surgery in Patients without Comorbidities"

_medicina, 2023, doi:10.3390/medicina59071196_

Round 1

Reviewer 1 Report

Respected authors 

My comments on your paper 

1- Retrospective study design with no aims and objectives regarding why the study was done 

2-There is no clear scientific message coming out of your studies

3- - There is no new finding coming out of your studies 

4- Numerous studies in the literature have established well accepted guidelines regarding Tourniquet use during knee arthroplasty in terms of indications, methods and also when not to use ..... so your study is not contributing anything.

5-The study design is flawed as there is no clear aims and objectives regarding on what research hypothesis this study was undertaken.

Thank you

English quality is average and needs no change .

Author Response

ANSWER Reviewer 1.

Dear Reviewer 1, we read your opinions and your own free thoughts. Even if you underestimate it, the data obtained from this article are the data obtained from the cases belonging to the Thrace region in Turkey. We definitely believe that this valuable information will contribute to the literature. But still, thank you for your time.

Kind regards!

Reviewer 2 Report

Dear authors,

thank you for submitting your work to the journal. 

I read it and gave my comments to the attached pdf-file. 

May major concerns are the fact that you did not present convincing data to support your conclusion. 

Kindly revise the paper completely.

The English language use is acceptable but could be improved. 

In some areas it is not clear what the authors message is. 

Kindly revise this. 

Author Response

ANSWER Reviewer 2.

Dear Reviewer 2, first of all, we would like to thank you for taking your time and making valuable contributions to our article. The article was reviewed and necessary revisions were made. I hope this reassures you. With our best regards.

Reviewer 3 Report

This study entitled “With or Without a Tourniquet? A Comparative Study on Total Knee Replacement Surgery in Patients without Comorbidities” seems to have been generally well executed and written. Furthermore, I believe that this paper will be of great interest to the readers. Finally, I have a few minor suggestions to further improve the quality of the paper.

Introduction

Insert the aim of study in the new paragraph. Furthermore, add the clear hypothesis of your study at the end of Introduction (i.e., in the same paragraph as the aim).

Materials and Methods

Ethical approval please insert following the first sentence in the Materials and Methods.

Add some subsections in this section of your paper.

Statistical analysis

Why the sample size calculation was not performed?

Discussion

Please begin this section with the main findings of your study.

Another limitation of your study represents the retrospective design so this issue must be discussed.

Author Response

Reviewer 3

This study entitled “With or Without a Tourniquet? A Comparative Study on Total Knee Replacement Surgery in Patients without Comorbidities” seems to have been generally well executed and written. Furthermore, I believe that this paper will be of great interest to the readers. Finally, I have a few minor suggestions to further improve the quality of the paper.

ANSWER Reviewer 3.

Thank you for your valuable contribution and positive comments. Upon your request, the article has been revised and necessary corrections have been made. With all due respect, we hope our revision has satisfied you.

Introduction

Insert the aim of study in the new paragraph. Furthermore, add the clear hypothesis of your study at the end of Introduction (i.e., in the same paragraph as the aim).

Sir, upon your request, the purpose of the research has been placed in a new paragraph and the explicit hypothesis of our study has been added at the end of the introduction.

Materials and Methods

Ethical approval please insert following the first sentence in the Materials and Methods.

Add some subsections in this section of your paper.

Sir, upon your request, ethical approval has been added after the first sentence of the Materials and Methods section, and some sub-headings have also been added to this section.

Statistical analysis

Why the sample size calculation was not performed?

Upon your request, the section about sample size has been added to the first part of statistical analysis.

Discussion

Please begin this section with the main findings of your study. Another limitation of your study represents the retrospective design so this issue must be discussed.

Dear Sir, all changes have been made upon your request, and both the discussion and conclusion sections have been revised. We would like to state it again, we would like to thank you for your positive contribution to our article. With my best regards.

Round 2

Reviewer 1 Report

RESPECTED AUTHORS

THE CORRECTIONS DONE ARE NOTED . 

NO COMMENTS